# Clinical Value of *IL6R* Gene Variants as Predictive Biomarkers for Toxicity to Tocilizumab in Patients with Rheumatoid Arthritis

**DOI:** 10.3390/jpm13010061

**Published:** 2022-12-28

**Authors:** Luis Sainz, Pau Riera, Patricia Moya, Sara Bernal, Jordi Casademont, Cesar Díaz-Torné, Ana Milena Millán, Hye Sang Park, Adriana Lasa, Héctor Corominas

**Affiliations:** 1Rheumatology Department, Hospital de la Santa Creu i Sant Pau (HSCSP), 08041 Barcelona, Spain; 2Faculty of Medicine, Universitat Autònoma de Barcelona (UAB), 08041 Barcelona, Spain; 3Institut d’Investigació Biomèdica Sant Pau (IIB SANT PAU), 08041 Barcelona, Spain; 4Pharmacy Department, Hospital de la Santa Creu i Sant Pau (HSCSP), 08025 Barcelona, Spain; 5CIBER de Enfermedades Raras (CIBERER), Instituto de Salud Carlos III, 28029 Madrid, Spain; 6Genetics Department, Hospital de la Santa Creu i Sant Pau (HSCSP), 08025 Barcelona, Spain; 7Medicine Department, Hospital de la Santa Creu i Sant Pau (HSCSP), 08025 Barcelona, Spain

**Keywords:** *IL6R*, genetic variants, tocilizumab, toxicity, rheumatoid arthritis, predictive biomarkers

## Abstract

Tocilizumab is a first-line biologic disease-modifying anti-rheumatic drug (bDMARD) that inhibits the interleukin-6 (IL-6) pathway by antagonizing the IL-6 receptor (IL-6R). Tocilizumab is widely used to treat rheumatoid arthritis (RA), a prevalent autoimmune disease that can cause irreversible joint damage and disability. Although many bDMARDs have been developed for RA, there is a lack of validated biomarkers which could guide personalized medicine strategies. To evaluate whether single-nucleotide polymorphisms (SNPs) in the *IL6R* gene could predict tocilizumab toxicity in patients with RA, we conducted a retrospective cohort study of 88 patients treated with tocilizumab. Six SNPs previously described in the *IL6R* gene were genotyped (rs12083537, rs11265618, rs4329505, rs2228145, rs4537545, and rs4845625). Using parametric tests, we studied the association between the SNPs and hepatotoxicity, infection, hypersensitivity, gastrointestinal, hematological, and dyslipidemia adverse events (AEs). We found associations between dyslipidemia and rs4845625 and between hematological AEs and rs11265618 and rs4329505. No further associations were found for the remaining SNPs and other AEs. Our findings support the potential clinical value of SNPs in the *IL6R* gene as predictive biomarkers for toxicity to tocilizumab in patients with RA.

## 1. Introduction

Rheumatoid arthritis (RA) is the most frequent autoimmune inflammatory disease worldwide, with an overall prevalence of 0.2–1% [1,2]. It is characterized by synovial inflammation, hyperplasia, and the progressive destruction of bone and cartilage in multiple joints [3], which, without appropriate treatment, may lead to progressive disability, systemic complications, and reduced life expectancy [4]. The current therapeutic strategy, consisting of early treatment based on a treat-to-target approach, aims to achieve remission or low disease activity and ultimately prevent structural damage [5,6,7,8,9,10].

At present, there is lack of evidence to recommend the treatment of one biologic disease-modifying anti-rheumatic drug (bDMARD) over another, as there are no validated biomarkers that predict response or toxicity to bDMARDs. Consequently, treatment selection is a trial-and-error process, and this can delay the identification of the most effective and well-tolerated treatment for a patient, which, in turn, may lead to disease progression and/or a poorer quality of life [11,12,13]. Although there is growing interest in the role of genetic markers as predictors of response/toxicity to biological drugs, most studies published to date have focused on methotrexate [14,15,16,17] and anti-tumor necrosis factor (anti-TNF) treatments [18,19,20,21,22,23,24].

Tocilizumab is a first-line bDMARD that competitively binds to both the membrane-bound and soluble forms of human interleukin-6 receptor (IL-6R), thereafter inhibiting the interleukin-6 (IL-6) pathway. It is an effective treatment for patients with moderate to severe RA, with reported response rates of 62–80% [13,25,26,27]. Despite being generally well-tolerated, approximately 80% of tocilizumab-treated patients experience adverse events (AEs) [28], most frequently infections, followed by gastrointestinal disorders. Although AEs are mainly mild or moderate, in 4% of patients, they are severe. To date, no biomarkers or predictors of toxicity have been identified.

Given that tocilizumab inhibits IL-6R, we hypothesized that functional variations in its encoding gene could affect tocilizumab toxicity. While previous studies have assessed the influence of *IL6R* gene variants on tocilizumab response, the results are inconsistent and not applicable to clinical practice [28,29,30,31,32,33]. As for toxicity, no studies to date have evaluated the influence of *IL6R* gene variants on tocilizumab-induced toxicities. The aim of this study was to assess whether genetic variants in the *IL6R* gene are associated with tocilizumab toxicity in patients with RA.

## 2. Materials and Methods

### 2.1. Study Population

We conducted a single-center retrospective cohort study including RA patients recruited in a tertiary referral hospital. The potential patients were identified using pharmacy registries in which we filtered for the patients who underwent tocilizumab treatment between January 2016 and January 2021.

We applied the following inclusion criteria: RA diagnosed according to the criteria of the American College of Rheumatology (ACR)/European League Against Rheumatism (EULAR) 2010 [34], prescription of tocilizumab for RA, and age over 18 years old. The exclusion criteria were other rheumatic diagnoses such as connective tissue diseases or vasculitis and loss of patient follow-up.

Sociodemographic and clinical data collected from electronic medical records were as follows: age, gender, age at diagnosis, previous treatments, comorbidities, body mass index, tocilizumab starting date, dose, and administration route, concomitant treatments, rheumatoid factor (RF), anti-citrullinated protein antibody (ACPA), reason for tocilizumab discontinuation, and AEs.

AEs were classified in the following groups: hepatotoxicity, infections, hypersensitivity, gastrointestinal, hematological, and dyslipidemia. For hepatotoxicity, hematological, and dyslipidemia AEs, periodic blood tests were revised retrospectively and, when an AE occurred, numeric values for bilirubinemia, transaminases, leukocytes and neutrophils, platelets, and lipids (total cholesterol, low-density lipoprotein (LDL), and triglycerides) were recorded. AEs were graded as mild (grade 1), moderate (grade 2), severe (grade 3), life-threatening (grade 4), and death (grade 5), in accordance with the Common Terminology Criteria for Adverse Events (CTCAEs) v6.0 [35].

Patients who discontinued tocilizumab without relation to AEs, death, or external causes were classified in the primary failure group, when discontinued in the first 6 months of treatment, and in the secondary failure group, when discontinued after a period of efficacy (>6 months).

The study was approved by the corresponding institutional ethics committees and registered at clinicaltrials.gov (protocol code: IIBSP-IIL-2020-148). All the participants gave written informed consent for blood sampling and genetic analyses.

### 2.2. Genetic Studies

Considering published data and bearing in mind single-nucleotide polymorphisms’ (SNPs’) functionality in the *IL6R*, we selected SNPs as follows: rs12083537, rs11265618, rs4329505, rs2228145, rs4537545, and rs4845625 (Table 1). The rationale for selecting these SNPs has been described in previous studies [28,29,30,31,36]. All the SNPs presented a minor allele frequency (MAF) over 0.10 in the European population according to the Allele Frequency Aggregator (ALFA) [37].

Genomic DNA was automatically extracted from peripheral whole-blood samples (Autopure, Qiagen, Hilden, Germany). The SNPs were analyzed via real-time polymerase chain reaction (PCR) using TaqMan SNP genotyping assays (Applied Biosystems, Foster City, CA, USA). All the cases were successfully genotyped.

### 2.3. Statistical Analysis

Hardy–Weinberg equilibrium was assessed for each analyzed SNP using the chi-square test (χ2). Codominant, dominant, and recessive models of inheritance were considered to assess associations with outcome variables whenever appropriate.

Quantitative data were expressed as mean (± SD) for normally distributed variables. Normality was assessed using the Shapiro–Wilks test. Depending on the number of groups compared, the Student’s *t*-test or ANOVA were used for normally distributed variables. Bivariate association between qualitative dichotomous variables was assessed using χ2 or Fisher’s exact test. The associations between SNPs and qualitative response variables were tested using the χ2 test.

All the tests were two-sided, significance was set to 5% (α = 0.05), and statistical analyses were performed using IBM-SPSS (IBM Corp. Released 2019. IBM SPSS Statistics for Windows, Version 26.0. Armonk, NY, USA: IBM Corp).

## 3. Results

### 3.1. Patient Characteristics

Table 2 summarizes the baseline demographic and clinical characteristics for the 88 patients that were included in the study. The mean disease duration on tocilizumab initiation was 16.5 (SD 11.9) years, and treatment survival was 45.91 (SD 40.1) months. Just over a third of patients (35.2%, *n* =31) were in active treatment at the end of data collection. The primary reason for tocilizumab discontinuation was AEs, followed closely by secondary and primary failure of treatment.

### 3.2. Adverse Events

Table 3 summarizes the frequency and severity of detected AEs, showing that hematological and dyslipidemia AEs were the most frequent. Most hematological AEs were mild and only required observation (*n* = 16) or dose reduction (*n* = 9). Neutropenia was the most common toxicity, with a median value of 1.17 (SD 0.03) ×10^9^/L; however, there were no severe cases, and tocilizumab was discontinued in only two patients. When thrombocytopenia occurred, the median value of platelets was 108.5 (SD 11.5) ×10^9^/L. As for dyslipidemia, the mean values of lipid values were as follows: total cholesterol, 284.5 (SD 42.0) mg/dL; LDL, 180.2 (SD 55.7) mg/dL; and triglycerides, 215.3 (SD 144.5) mg/dL. Regarding dyslipidemia management, 15 patients were initiated on hypolipemiant treatment during tocilizumab treatment.

The third most frequent AE was hepatotoxicity, with more cases of hypertransaminasemia (mean aspartate aminotransferase (AST), 62.7 (SD 21.2) IU/L and alanine aminotransferase (ALT), 92.1 (SD 42.2) IU/L) than of hyperbilirubinemia (mean bilirubin, 24.9 (SD 14.4) µmol/L). Only seven hepatotoxicity AEs were graded as moderate; four of those cases developed during concomitant methotrexate treatment and were managed effectively with methotrexate dose reduction.

The most severe AEs were infections, with three patients requiring hospitalization; all the cases were managed with temporary tocilizumab suspension. All seven patients showing hypersensibility to tocilizumab were managed with permanent discontinuation.

### 3.3. Genetic Determinants and Adverse Events

The genotypic frequencies of the six studied SNPs did not significantly deviate from the expected Hardy–Weinberg equilibrium. The mutated allele (C) of the SNP rs4845625 was more frequent than the wild-type (T) allele, as expected from allele frequency population-based studies [37].

In the univariate analysis, three SNPs (rs4845625, rs1126518, and rs432955) showed statistically significant associations with hepatotoxicity, hematological, and dyslipidemia AEs (Table 4). The CC carriers for rs4845625 developed dyslipidemia less frequently (16.7%) than the patients carrying the T allele (TT + CT) (36.7%) (*p* = 0.04). Considering a codominant model for rs4845625 and hepatotoxicity, statistically significant differences were found between the three genotypes (*p* = 0.048). Patients carrying the CC genotype for rs11265618 showed a higher rate of hematological AEs compared to T carriers (CT + TT) (36.7% vs. 14.3%, *p* =0.032). Comparably, the TT genotype for rs4329505 also demonstrated more hematological AEs than C carriers (CT + CC) (36.1% vs. 14.8%, *p* = 0.044).

Further statistically significant associations were not found, either for the remaining AEs detected during tocilizumab treatment or for the remaining SNPs (rs12083537, rs2228145, and rs4537545).

## 4. Discussion

We assessed associations between six SNPs in the IL6R gene and AEs during tocilizumab treatment in 88 patients with RA. We found an association between rs4845625 and dyslipidemia rates and associations between rs11265618 and rs4329505 and hematological AEs.

Previous studies focused on *UGT1A1* polymorphisms and hyperbilirubinemia have reported results for Japanese patients with RA [38,39]. However, this is the first study demonstrating that SNPs in the *IL6R* gene may influence the development of AEs during long-term treatment with tocilizumab in patients with RA.

Our results regarding rs4845625 show that carriers of the T allele (TT + TC) had a higher incidence of dyslipidemia compared to CC carriers (36.7% vs. 16.7%) when treated with tocilizumab. A recent study of patients with RA by our group also described a greater response to tocilizumab treatment at six months in carriers of the TT or CT genotypes for rs4845625 [33]. Additionally, in line with our results, in 2017, Arguinano et al. described an association between rs4845625*T and high levels of LDL in a healthy cohort, suggesting that *IL6R* genetic variations may constitute a cardiovascular risk factor [36]. Although it seems that genotypes associated with better efficacy outcomes are also related to greater rates of dyslipidemia, this should not be problematic from a treatment biomarker perspective, given that dyslipidemia secondary to tocilizumab treatment has not been linked to cardiovascular events [40,41,42,43,44,45]. In fact, the association between the inflammatory burden in RA with changes in the lipid profile and cardiovascular risk has been described as the “lipid paradox”, as lower cholesterol is associated with higher cardiovascular risk in RA untreated patients [46]. Although the mechanism underlying tocilizumab effects on the lipid profile is unclear, it has been demonstrated that, in a proinflammatory state with the activation of IL-6 pathways, there is a lowering of serum lipids mediated by the modification of apolipoprotein synthesis [47] as well as IL-6 local effects on fat tissue and muscle [44]. It has also been hypothesized that tocilizumab treatment raises lipid levels by reversing the lowering of lipid levels induced by IL-6-mediated inflammation [47,48]. Another suggested explanation is that dyslipidemia is mediated by the direct effect of IL-6 levels, raised through the blockage of IL-6R on administering tocilizumab [44].

We also observed an increase in compounding hematological AEs for the CC genotype for rs11265618 and for the TT genotype for rs4329505. In previous research, these genotypes have been described as potential biomarkers to guide tocilizumab treatment in RA cohorts. CC carriers of rs11265618 showed improved low disease activity rates at 12 months [28], while TT carriers of rs4329505 showed less joint swelling at three months [30]. Although we found similar trends relating to the same genotypes with efficacy outcomes at six months in our previous study, the results were not statistically significant [33].

We found no association between the rs12083537, rs2228145, and rs4537545 genetic variants and AEs detected during tocilizumab treatment. The rs12083537 variant has previously been studied, with conflicting results; while Enevold et al. [30] found that carriers of the AA genotype presented a poorer response to tocilizumab treatment, other studies found the opposite [29,31], and a recent study found no association [33]. As for the *IL6R* SNP rs2228145, this has only been proposed as a biomarker of response to tocilizumab by Enevold et al. [30]. On the other hand, Arguinano et al. [36] have reported rs4537545*C to be associated with higher LDL levels in a cohort of healthy patients, although only when considered as a haplotype with rs4845625*T. In our research, we did not consider haplotypes, and we did not find increased dyslipidemia for the rs4537545 genetic variant.

The main limitation of this study is its retrospective design. The relatively high incidence of AEs reported is possibly explained by our exhaustive search for hepatic, hematologic, and lipid profile abnormalities in routine blood tests over long periods of tocilizumab treatment. This would also explain the predominance of mild AEs that did not require tocilizumab discontinuation. Although the homogeneity of our cohort and the relatively large sample size reinforce the validity of our results, further prospective studies are warranted before the implementation of our findings in clinical practice.

## 5. Conclusions

Our *IL6R* polymorphism results suggest that, during the treatment of patients with active RA with tocilizumab, rs4845625 is associated with the development of dyslipidemia, and rs11255618 and rs4329505 are associated with the development of hematological effects.

## Figures and Tables

**Table 1 jpm-13-00061-t001:** Selected functional polymorphisms in the *IL6R* gene.

Genetic Variant	Genomic Position (GRCh38)	MAF	Alleles
rs12083537	chr1:154408627	0.21	A>**G**
rs11265618	chr1:154457616	0.17	C>**T**
rs4329505	chr1:154459944	0.17	T>**C**
rs2228145	chr1:154454494	0.40	A>**C**
rs4537545	chr1:154446403	0.41	C>**T**
rs4845625	chr1:154449591	0.43	**T**>C

Abbreviations: *IL6R*, interleukin-6 receptor. MAF, minor allele frequency reported in the European population. The minor allele is highlighted in bold.

**Table 2 jpm-13-00061-t002:** Patient baseline characteristics (*n* = 88).

Variables	*n* (%)	Mean (SD)
Female/male	76 (86.4) / 12 (13.6)	
Age on diagnosis (years)		46.6 (15.9)
Disease duration (years)		16.5 (11.9)
RF positive	58 (65.9)	
ACPA positive	57 (64.8)	
Body mass index		28.6 (6.1)
Number of previous cDMARDs		2.4 (1.3)
Number of previous bDMARDs		1.5 (1.4)
Tocilizumab administration (intravenous)	47 (53.4)	
Treatment survival (months)		45.91 (40.1)
Reasons for discontinuation (*n* =57)		
Primary failure	14 (15.9)	
Secondary failure	16 (18.2)	
AE	19 (21.6)	
Unrelated death	1 (1.1)	
Other	7 (8.0)	

Abbreviations: ACPA, anti-citrullinated protein antibodies; AE, adverse event; bDMARD, biological disease-modifying anti-rheumatic drug; cDMARD, conventional disease-modifying anti-rheumatic drug; RF, rheumatoid factor; SD, standard deviation.

**Table 3 jpm-13-00061-t003:** Reported adverse events (*n* = 88).

Variables	*n* (%)	Mild *n* (%)	Moderate *n* (%)	Severe *n* (%)
Hepatotoxicity	20 (22.7)	13 (14.7)	7 (7.9)	0
Hyperbilirubinemia	7 (7.9)			
Hypertransaminasemia	13 (14.7)			
Treatment discontinuation	5 (5.7)			
Infections	18 (20.5)	9 (10.2)	6 (6.8)	3 (3.4)
Tocilizumab discontinuation	1			
Hypersensitivity	7 (7.9)	0	7 (7.9)	0
Tocilizumab discontinuation	7			
Gastrointestinal	3 (3.4)	1 (1.1)	2 (2.3)	0
Tocilizumab discontinuation	0			
Hematological	27 (30.7)	19 (21.6)	8 (9.1)	0
Neutropenia	23 (26.1)			
Thrombocytopenia	6 (6.8)			
Tocilizumab discontinuation	2 (2.3)			
Dyslipidemia	27 (30.7)			
Statin initiation	15 (17)			

**Table 4 jpm-13-00061-t004:** Associations between rs4845625, rs11265618, and rs4329505 and hepatotoxicity, hematological, and dyslipidemia AEs.

SNPs	Genotype (*n*)	Hepatotoxicity AEs				Hematological AEs				Dyslipidemia AEs			
		%	Genetic Model	OR (95% CI)	*p*	%	Genetic Model	OR (95% CI)	*p*	%	Genetic Model	OR (95% CI)	*p*
rs4845625	T/T (13)	7.7	Cod	-	0.048	38.5	Cod	-	0.712	23.1	Cod	-	0.051
	C/T (45)	33.3	Dom	4.07 (0.49–33.42)	0.161	26.7	Rec	1.03 (0.39–2.71)	0.946	42.2	Rec	3.05 (1.02 -9.17)	0.04
	C/C (30)	13.3				30				16.7			
rs11265618	C/C (60)	16	Cod	-	0.603	36.7	Cod	-	0.075	33.3	Cod	-	0.447
	C/T (25)	25	Dom	0.65 (0.21–2.02)	0.456	12	Dom	3.47 (1.07–11.36)	0.032	28	Dom	0.67 (0.24–1.83)	0.43
	T/T (3)	33.3				33.3				0			
rs4329505	T/T (61)	24.6	Cod	-	0.665	36.1	Cod	-	0.099	32.8	Cod		0.388
	C/T (24)	16.7	Dom	0.69 (0.23–2.16)	0.531	12.5	Dom	3.25 (1–10.64)	0.044	29.2	Dom	0.72 (0.26–1.98)	0.52
	C/C (3)	33				33.3				0			

Abbreviations: Cod: codominant; Dom: dominant; OR: odds ratio; Rec: recessive; SNP: single-nucleotide polymorphism.

## Data Availability

The data presented in this study are available upon request from the corresponding author. The data are not publicly available due to the inclusion of clinical and personal data.

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
