# Peer review of "Clinical Value of IL6R Gene Variants as Predictive Biomarkers for Toxicity to Tocilizumab in Patients with Rheumatoid Arthritis"

_jpm, 2022, doi:10.3390/jpm13010061_

Round 1

Reviewer 1 Report

Dear author,

an outstanding idea and work. 

Good, clearly written English with an important conclusions  of importance for future every day work. 

Kind regards,

Dragana

Author Response

Ms. Agnes Cao

Assistant Editor

Journal of Personalized Medicine

Barcelona, 22nd December 2022

Dear Dr. Dragana,  

We would like to take this opportunity to thank you for your valuable comments and your evaluation of our manuscript.

Yours sincerely,

Pau Riera, PharmD, PhD                      Héctor Corominas, MD, PhD         

Pharmacy Department                          Rheumatology Department

Hospital de la Santa Creu i Sant Pau

Carrer Sant Quintí, 89, 08041, Barcelona, Spain

Reviewer 2 Report

1. Reference management is suboptimal: for example, the first 2 references appear in the text as “[1], [2]”, when they should probably look like this: “[1,2]”; similarly, references “[5]–[10]” should probably be “[5–10]” etc.

2.   A question of style, the author may proceed as they desire, but abbreviating a single word is unnecessary in the body text of the manuscript (for example using “MTX” for “methotrexate” or “TCZ” for “tocilizumab”), since it does not reduce the word count and it fills up the text with a lot of capital letters making it tiresome to read. It would make sense abbreviating single words in a table or in a figure for lack of typing space.

3.      “2.1. Study Population”: the authors write that they included “88 RA patients”. This is a result not a method: the methods section of the article describes how the authors included patients, and the results section of the article reports how many they included using the specific method declared previously.

4.      “2.1. Study Population”: the authors write that they included “88 RA patients” between “2016 and 2021”. This number of patients is rather low for a 6-year study interval. This should be explained.

5.      “2.1. Study Population”: where did the authors get their RA patients? Were they from an out-clinic, tertiary rheumatology center, clinical study etc.? Were they hospital in-patients? Did they come from a single center or from multiple centers? These details are significant for patient profile.

6.      “2.1. Study Population”: how do the authors know these patients had RA? By ICD codes? Did their RA diagnosis fulfill the current RA classification criteria?

7.      “2.1. Study Population”: the authors state that they included patients between “2016 and 2021”. The interval is vague; at least the months should be reported.

8.      “2.1. Study Population”: the authors state that they included patients between “2016 and 2021”. Were the patients included randomly? How?

9.      “2.1. Study Population”: the authors state that they included patients between “2016 and 2021”. What were the inclusion criteria?

10.  “2.1. Study Population”: the authors state that they included patients between “2016 and 2021”. What were the exclusion criteria?

11.  “2.1. Study Population”: the authors stated that they collected “age, gender … etc.”. Did they not collect efficacy data? Such as composite activity scores?

12.  “2.1. Study Population”: the authors state the study was approved by their ethics committee. But we do not read the fact that the patients gave written informed consent. This was mandatory for both data usage for scientific purposes and genetic testing of their blood.

13.  “2.1. Study Population”: patients were categorized “in the primary failure group” and in “the secondary failure group”. How was failure defined?

14.  “2.2. Genetic Studies”: the authors stated that “genomic DNA was automatically extracted from peripheral whole-blood samples”. Where did they have the blood from, since this was a retrospective study? Was it stored? Why?

15.  The authors write “Statistical AnalysEs”. They mean “Statistical AnalysIs”.

16.  The heading “Statistical Analyses” is not numbered. It should be “2.3. Statistical Analysis”.

17.  “Statistical Analysis”: SPSS is not properly cited, please see: https://www.ibm.com/support/pages/how-cite-ibm-spss-statistics-or-earlier-versions-spss.

18.  “Statistical Analysis”: the authors declare that “bivariate association between qualitative dichotomous variables was assessed 106 using Pearson’s χ2”. They meant “Pearson’s r” or “Pearson’s correlation coefficient”.

19.  “Results”: Table 2 contains “erosive RA” variable. This was not mentioned in the Methods section. How was is defined?

20.  “Results”: Table 2 contains “body mass index”. This was not mentioned in the Methods section.

21.  “Results”: section “3.2. Adverse Events” reports laboratory variables (complete blood counts, total cholesterol and fractions, transaminases, bilirubin) that were not mentioned in the Methods section.

22.  “Results”: Table 4 is hard to follow. Please a lower font size and adjust cell size so that all rows align.

23. “Discussion”: the authors write that, “as far as we are aware, his is the first study demonstrating that SNPs in the IL6R gene may influence the 173 development of AEs during long-term treatment with TCZ in patients with RA”. A cautious mode of expression is wise (“as far as we are aware”), but the authors need to be more confident that their study is the only one that does what it does. This comes from a reasonable literature search (e. g. PubMed, SCOPUS, Web of Science).
